# Hepatic and Skeletal Muscle Autophagy Marker Levels in Rat Models of Prenatal and Postnatal Protein Restriction

**DOI:** 10.3390/nu15133058

**Published:** 2023-07-07

**Authors:** Irena Santosa, Hiromichi Shoji, Yoshiteru Arai, Kentaro Awata, Kazuhide Tokita, Toshiaki Shimizu

**Affiliations:** 1Department of Pediatrics and Adolescent Medicine, Graduate School of Medicine, Juntendo University, Bunkyo, Tokyo 113-8421, Japan; s-irena@juntendo.ac.jp (I.S.); tshimizu@juntendo.ac.jp (T.S.); 2Department of Pediatrics, Faculty of Medicine, Juntendo University, Bunkyo, Tokyo 113-8421, Japan; yo-arai@juntendo.ac.jp (Y.A.); k-awata@juntendo.ac.jp (K.A.); ka-tokita@juntendo.ac.jp (K.T.)

**Keywords:** fetal growth restriction, autophagy, protein restriction, maternal nutrition, insulin resistance

## Abstract

Fetal growth restriction (FGR) leads to adult-onset metabolic syndrome. Intrauterine and early postnatal caloric restriction ameliorates the risk in animal models. To understand the underlying mechanism, we compared autophagic marker levels between offspring with FGR and those with prenatal and early postnatal protein restriction (IPPR). We postulated that FGR would impair, whereas IPPR would help regulate, autophagy in neonatal rats. This study involved control (Con), FGR offspring (Pre), and IPPR offspring groups (Pre + Post); *n* = 5/group. We assessed the abundance of autophagy markers in the liver and skeletal muscles. At birth, the Pre group pups had lower levels of some autophagy-related proteins, with increased p62 expression and a low microtubule-associated protein light chain beta (LC3-II:LC3-I) ratio. This finding suggests a lower hepatic autophagy flux in FGR offspring than the Con group. The hepatic levels of autophagy proteins were considerably decreased in the Pre and Pre + Post groups at 21 days of age compared to the Con group, but the LC3-II:LC3-I ratio was higher in the Pre + Post group than in the Con and Pre groups. The muscle levels of beclin-1, LC3-II, and p62 were lower in the Pre group pups, with no difference in the LC3-II:LC3-I ratio among the groups. An imbalance in the nutritional environment is associated with downstream autophagic flux, thus suggesting that FGR offspring will have impaired autophagic flux, and that post-natal nutrition restriction might help reduce this risk.

## 1. Introduction

Maternal undernutrition has become a concern in Japan as it leads to high rates of fetal growth restriction (FGR), which affects 3–7% of all pregnancies and is the second leading cause of death in neonates after prematurity [1]. FGR is defined as impaired fetal development due to uteroplacental insufficiency, in which the placenta does not fulfill the metabolic requirements based on the fetal genome. Fetal development then stops for several weeks. The causes of FGR include inappropriate substrate availability due to maternal undernutrition, drug use, and infection [2,3], and FGR results in decreased body weight at birth. The prevalence of low birth weight (LBW) has increased from 5.2% in 1980 to 9.4% in 2021. The mean birth weight in Japan has also significantly decreased from 3230 and 3140 g in 1980 to 3050 and 2960 g in 2021 among male and female individuals, respectively [4]. Observational and experimental studies have shown that the first trimester of gestation is the most vulnerable period of fetal growth [5,6]. 

The fetal origin of the adult disease hypothesis propounds that chronic disease onset in adults is nascent in utero. The weight of a baby at birth and long-term metabolic health correlate inversely, as in utero adverse events might influence metabolic functions in organs permanently. Infants with FGR have impaired organ growth with metabolic diseases, such as type 2 diabetes (DM), obesity, cardiovascular disease, and hypertension in adulthood [7]. The mechanism through which FGR leads to metabolic syndrome remains poorly understood. In this milieu, studies have investigated the causes of FGR as well as the metabolic pathologies that arise in response to impaired organ development [8]. Reduced insulin sensitivity is the basic pathological rationalization for the susceptibility of children with FGR to DM and cardiovascular disease in adulthood, in line with the “Barker Hypothesis” or “Developmental Origins of Health and Disease” (DOHaD) [7]. Recently, the theory that autophagy is one of the factors in insulin resistance has drawn our attention. 

Autophagy is a highly regulated cell death process that is involved in the clearance of damaged organelles, long-lived proteins, and intracellular pathogens, and it is a degradative pathway in all eukaryotic cells. Catabolic processes are involved in maintaining cellular homeostasis by transferring cytoplasmic components to the lysosomes for degradation [9]. Autophagy helps maintain intracellular homeostasis by regulating the intrinsic metabolism of amino acids or other nutrients [10]. Microtubule-associated protein 1 light chain beta (LC3-I) is converted to LC3-II via lipidation through a ubiquitin-like system involving autophagy-related gene 5 (Atg5). During this process, LC3-II is incorporated into an expanding autophagosomal membrane, which functions as a protein that binds to adapter proteins, such as sequestosome (p62), that engage components in cells for breakdown. Autophagy is a highly dynamic and multi-step process [11]. 

Autophagy impairments are associated with various diseases, including DM, neoplasm, neurodegeneration, and infectious diseases, as well as aging. Autophagy dysregulation in the pancreas and liver has been associated with obesity and DM, which explains why autophagy has attracted interest in DOHaD [12]. The expression of Atg genes (essential genes for autophagosome formation) and the proteins is altered in the skeletal muscle, liver, and adipose tissue in insulin-resistant humans, as well as in mouse models of diet-induced or genetic obesity [13,14]. Endoplasmic reticulum (ER) stress can induce autophagy in mammalian cells, probably through the degradation of unfolded proteins [15]. The mechanism underlying this process is heterotopic fat accumulation—after a certain level of accumulation in adipose tissue, fat accumulates in non-adipose tissue, such as the liver and skeletal muscle. Ectopic fat accumulation might result in lower insulin sensitivity [16]. Therefore, we decided to investigate the autophagy level in the liver and skeletal muscle.

Rats with intrauterine and postnatal caloric restriction are protected against long-term adult sequelae [7]. This finding indicated that to develop postnatal therapeutic measures to reduce the risk of metabolic syndrome in adults, it is necessary to apprehend the origins of cell stress, autophagy, and programmed cell death in FGR offspring. A rapid postnatal weight gain or catch-up growth in LBW neonates can indirectly lead to cellular stress and metabolic disease. Although autophagy impairments in beta cells of the pancreas or insulin target tissues have been investigated, the metabolic effect of autophagy in the liver and skeletal muscle has not been compared between Prenatal and Pre + Postnatal protein-restricted neonates. In this study, we investigated whether FGR offspring have autophagic impairments, and whether this impairment can be prevented by keeping the same nutrients as intrauterine nutrients. 

## 2. Materials and Methods

The Animal Care Committee of Juntendo University, Tokyo, Japan approved the study protocol (approval no. 1494). Rats were anesthetized using isoflurane to minimize suffering.

### 2.1. Animals

Eight- to ten-week-old pregnant Sprague–Dawley (SD) rats (Sankyo Labo Service Corporation, Inc., Tokyo, Japan) were housed individually in cages with ad libitum access to food and water, under a 12 h light/12 h dark cycle at 24–25 °C, until the rats spontaneously delivered their litters. Five neonates closest to the median weight in each group were culled, and the remaining neonates were sacrificed to harvest the livers. The dams nursed the neonates until weaning at 3 weeks, when all pups were sacrificed. Blood glucose was measured before the sacrifice with Precision Xceed (Cat. No. 71085-80; ABBOTT Japan, Chiba, Japan). Liver and skeletal muscle samples were collected from the posterior compartment of the hind legs. The samples were snap-frozen in liquid nitrogen and stored at −80 °C until analysis.

### 2.2. Prenatal and Postnatal Nutrition Management

There were three groups in this study. (1) Control-fed dams (Con, *n* = 5) were allowed free access to food throughout the gestational period until parturition day 21 through lactation for 3 weeks. (2) The rats exposed to intrauterine protein restriction (Pre, *n* = 5) were born to mothers fed isocaloric low protein (LP) chow (7% protein, 66.8% carbohydrate, and 6.5% fat; Nihon Clea Inc., Tokyo, Japan) from gestational day 1 to birth (day 21), and because of the maternal nutrient restriction, the pups were born with FGR and LBW. After birth, the offspring were allowed ad libitum access to breast milk from dams with standard chow (21% protein, 52.7% carbohydrate, and 6.6% fat; Nihon Clea Inc.). (3) The third group was predisposed to combined intrauterine and postnatal protein restriction (Pre + Post, *n* = 5); the pups were born to mothers exposed to LP chow from gestational day 1 to day 21, and after birth, the dams were maintained on LP chow.

### 2.3. Western Blotting

Hepatic and skeletal muscle tissues were homogenized in a vibratory crusher with RIPA buffer containing 50 mM Tris-HCl (pH 7.6), 150 mmol/L NaCl, 1% Nonidet^®^ P40, 0.5% sodium deoxycholate, Protease Inhibitor Cocktail, and 0.1% SDS (Cat. No. 08714-04; Nacalai Tesque, Kyoto, Japan). The Pierce™ BCA Protein Assay Kit (Cat. No. 23225; Thermo Fisher Scientific, Inc., Danvers, Waltham, MA, USA) was used to quantify the proteins. The proteins (10 μg/sample) were resolved using NuPAGE™ LDS Sample buffer (Thermo Fisher Scientific Inc.) and then transferred onto polyvinylidene difluoride (PVDF) membranes. Non-specific antigen binding was blocked with Bullet Blocking One for Western Blotting (Cat. No. 13779-56; Nacalai Tesque) for 10 min; then, the samples were incubated with beclin-1 (1:1000; Cat. no. 3738; Cell Signaling Technology, Danvers, MA, USA), Atg12 (1:1000; Cat. no. 4180; Cell Signaling Technology), LC3I/II (1:1000; Cat. no. 12741; Cell Signaling Technology), p62 (1:1000; Cat. no. 5114; Cell Signaling Technology), CHOP (1:1000; Cat. no. 2895; Cell Signaling Technology), Anti-GRP78 BiP (1:1000; ab21685; Abcam; Cambridge, UK), and glyceraldehyde-3-phosphate dehydrogenase (GAPDH) (1:1000; Cat. no. 5174s; Cell Signaling Technology) antibodies at 4 °C overnight. GAPDH was used as the internal reference. After washing 3 times with Tris-buffered saline with 0.1% Tween-20 (TBST), the samples were gently shaken with horseradish peroxidase (HRP)-conjugated goat anti-rabbit IgG (1:10.000; Cat. no. 7076; Cell Signaling Technology) for 1 h at 24 °C. After 4 washes with Tris-Buffered Saline with Tween 20% (TBST), the blots were visualized using ImmunoStar LD (Cat. no. 296-69901; FUJIFILM Wako Pure Chemical Corporation, Osaka, Japan). Fusion software ver. 1.51 was used for quantifying the band intensities (National Institute of Health, Bethesda, MD, USA).

### 2.4. Statistical Analysis

Data are expressed as mean ± SD. Data from day 0 (at birth) and week 3 were statistically analyzed using Student’s *t*-test and a one-way ANOVA, respectively. All data were analyzed using Prism Ver. 9 (GraphPad Software Inc., San Diego, CA, USA). Results with *p* < 0.05 were considered statistically significant.

## 3. Results

### 3.1. Body Weight and Glucose Level of the Rat Model

The Pre and Pre + Post pups weighed significantly less than the Con pups on day 0 (5.0 ± 0.7 vs. 7.1 ± 0.6 g, *p* < 0.0001). These results confirmed the establishment of the FGR model. After switching the maternal diet to standard chow for 1 week, the Pre pups weighed significantly more than the Pre + Post pups that were maintained on the LP diet (11.0 ± 0.2 vs. 8.4 ± 0.8 g). However, the Con pups still weighed significantly more than the other two groups’ pups (18.1 ± 0.7 g; Figure 1).

The growth rate of the Pre pups was faster than the Pre + Post pups’ by week 2 (25.4 ± 0.3 vs. 12.0 ± 2.3 g, *p* < 0.0001). By week 3, the Pre group pups had undergone “partial catch-up growth”, but the body weight still significantly differed from that of the Con pups (*p* = 0.01). The Pre + Post pups that were postnatally maintained on the LP diet did not show catch-up growth and remained small relative to the Pre and Con pups (14.2 ± 2.8 vs. 44.7 ± 1.0 and 49.1 ± 1.9 g, respectively, Figure 1).

The glucose level of the Pre + Post group (115.6 ± 8.649 mg/dL) was significantly lower than that of both the Con and Pre groups (158.4 ± 10.24 mg/dL; *p* = 0.0021 and 173.8 ± 22.74 mg/dL; *p* = 0.0002, respectively, Figure 2).

### 3.2. Hepatic and Skeletal Muscle Autophagy-Related Proteins

We evaluated the effects of protein restriction and low birth weight on autophagy. We quantified the liver and skeletal muscle protein expression—beclin-1 (autophagy initiation protein), prolongation protein Atg12, LC3-I, LC3-II, and termination phase protein p62—using Western blotting.

As shown in Figure 3, the hepatic protein levels of beclin-1 and LC3-I did not significantly differ among the Pre, Pre + Post, and Con neonates on day 0. The levels of the autophagy regulator protein Atg12 and LC3-II were considerably reduced (*p* = 0.0015 and *p* = 0.1, respectively), whereas the level of p62 was significantly increased (*p* = 0.024) in the livers of Pre group pups on day 0 compared to the Con group. 

By week 3, the hepatic protein levels of LC3-I and LC3-II did not significantly differ between the Pre and Pre + Post groups, but they were lower than those in the Con pups (*p* < 0.0001; Figure 4). In contrast, the hepatic level of beclin-1 increased in the Pre + Post group compared with that in the Pre group (*p* = 0.0064), but it did not significantly differ from the level in the Con group. The levels of Atg12 and p62 did not significantly differ between the groups.

We calculated the LC3-II:LC3-I ratio as one of the autophagy markers. The ratio in the livers of the Pre group on day 0 was lower, and the ratio of the Pre + Post pups at week 3 was higher than that of the Pre and Con pups (Figure 5).

As shown in Figure 6, the beclin-1, LC3-II, and p62 levels in the skeletal muscles of the Pre group significantly decreased compared with those in the skeletal muscles of the Con pups (*p* = 0.04, *p* = 0.04, and *p* = 0.01, respectively). The levels of Atg12, LC3-II, and p62 were also significantly higher (*p* = 0.02, *p* = 0.03, and *p* = 0.01, respectively) in the Pre + Post group than in the other groups. The levels of skeletal muscle autophagy regulator proteins did not significantly differ between the Pre and Pre + Post pups. The LC3-II:LC3-I ratio also did not significantly differ among the three groups.

### 3.3. Stress Markers in the Liver and Skeletal Muscle

We measured the levels of the stress markers CHOP and Bip (Anti-GRP78) in the liver on day 0 and at week 3 and in the skeletal muscle. There were no significant differences among the groups in either the liver (Figure 7) or the skeletal muscle (Figure 8).

## 4. Discussion

To the best of our knowledge, this is the first study to compare autophagy levels among prenatal nutrition-restricted offspring, prenatal and postnatal nutrition-restricted offspring, and control offspring. The majority of FGR newborns show catch-up growth during the first few years of life [17]. In this study, the Pre group pups also showed some “catch-up growth” by week 3, which is equal to 2 years of human age. However, the body weight was still significantly lower than that of the Con group. A previous study has shown that LBW offspring with early postnatal catch-up growth develop obesity and glucose intolerance, and suppressing early catch-up growth can prevent the development of glucose intolerance and obesity [18]. In this study, the Pre group showed some early postnatal “catch-up growth”, and the glucose level of this group was not significantly different from that of the Con group, but these pups may develop glucose tolerance in their adult life. In contrast, in the Pre + Post group pups, which did not exhibit “catch-up growth”, the glucose level was significantly lower than that in the Con and Pre groups. These results indicate that postnatal nutrition restriction may lower the risk of developing glucose intolerance in adult life. High glucose levels indicate high insulin levels, and insulin has been shown to inhibit the autophagy process. Exposure to high insulin levels for long periods may inhibit autophagy and thereby induce insulin resistance. 

In glucose metabolism, autophagy in the liver, muscle, heart, pancreas, and skeletal muscle plays an important role and utilizes approximately 80% of the glucose to mediate insulin action [19]. The liver is crucial for regulating whole-body glucose metabolism by maintaining the balance between glucose storage and release. As the liver is the major organ that manufactures and supplies blood glucose, we strived to elucidate the potential role of autophagy in the liver in reprogramming glucose metabolism disturbance. The results of studies on the liver and isolated hepatocytes have greatly contributed to understanding the importance of autophagy in metabolism. The key role of autophagy in hepatocytes is lipophagy, which is the process of selectively degrading cytoplasmic lipid droplets by the formation of autophagosomes, degradation of some lipid droplets, and then fusion with lysosomes [20]. 

Autophagy affects the levels of AMPK and mTORC1, which are upstream regulators of autophagy [21]. Autophagic flux refers to a series of processes in which the degradation occurs from the initiation of autophagy. In this initiation stage, beclin-1 plays a role in autophagosome formation. Under nutrient-deprived conditions, such as post-natal fasting, glucagon binds to its receptors in the liver, leading to the activation of AMPK and the inactivation of mTORC1, which stimulates autophagy [22]. 

Autophagy is a physiological mechanism associated with the degradation of intracellular components for energy production, building block recycling, and tissue remodeling. Autophagy is initiated by the formation of vesicular double-membrane autophagosomes that surround cytosolic cargo and fuse with the lysosomes, which digest the internalized contents. These steps are controlled by a series of Atg proteins and protein complexes, such as LC3, p62, beclin-1, Atg2, and the ULK1, PI3K, and Atg12 complexes [10]. The changes in autophagy protein levels in the process play an important role in the development of insulin resistance and various metabolic diseases [16]. 

Perinatal nutritional status plays a critical role in regulating autophagy in the livers of rats with FGR (day 0). An increased number of autophagosomes suggests either enhanced LC3B formation or reduced lysosomal clearance. The levels of hepatic markers of autophagy decrease in neonatal rats with protein restriction during gestation, and this results in increased protein polyubiquitination [22]. In this study, we found significantly lower levels of the autophagy prolongation proteins Atg12 and LC3-II, resulting in increased protein levels of p62. These results support the results in a type 2 DM mouse model, which showed a decrease in the LC3-II:LC3-I ratio, a primary marker of autophagic activity, and an increase in the p62 level. This result supported the findings of a previous study; that is, that impaired autophagy in hepatocytes is correlated with decreased lysosomal degradation rate, causing aggravation of ER stress to a greater extent. Inhibition of autophagy will lead to the accumulation of harmful substances in the cells and cause mitochondrial oxidative stress and ER stress, resulting in insulin resistance [16]. As observed in this study, low autophagy levels in the liver resulted in a tendency of high levels of ER stress markers CHOP and BiP, even though the differences in the levels were not significant.

We analyzed the p62 level to determine whether the lower expression of LC3-II was caused by reduced autophagosome formation or increased clearance. Accumulated p62 protein confirmed impaired autophagic flux. Decreased numbers of autophagosomes resulted in an increased p62 level in this study. Sequestosome 1 labels ubiquitin on target proteins and is degraded by lysosomes; it is an autophagy degradation marker. Impaired autophagy results in an increase in polyubiquitinated proteins through reduced clearance, which can generally lead to oxidative stress [14]. 

When autophagy is activated, the ratio of LC3-II:LC3-I increases, and vice versa [23]. In this study, the LC3-II:LC3-I ratio was relatively low in the Pre group. This finding suggested that hepatic autophagy was suppressed on day 0. Decreased Atg12 and LC3-II levels and increased p62 level indicated hepatic autophagy suppression on day 0, suggesting that FGR offspring have hepatic autophagic impairments that can result in insulin resistance later in life. Suppression of hepatic autophagy compromises the macromolecule rate of replacement, including glycogen and lipid turnover, causing insulin resistance [22]. 

Autophagy is essential for cytoplasmic recombination and mitochondrial clearance during fat formation. On the contrary, autophagy may be involved in the regulation of mitochondrial metabolism in mature adipocytes. Therefore, autophagy has a key role in adipocyte differentiation and lipid droplet accumulation [16]. As reported by a study from our laboratory, the levels of intracellular lipid droplets in an FGR rat model at 12 weeks were higher than those in the Con group, suggesting a compromised lipid turnover. Furthermore, the same study found decreased mRNA and protein expression of GLUT4 insulin signal in the soleus muscle, suggesting a relationship between impaired lipid turnover and insulin resistance in FGR rats [24].

In this study, the expression of beclin-1, LC3-II, and p62 decreased in the skeletal muscles of the Pre pups compared with that in the Con group. The levels of Atg12, LC3-II, and p62 were also significantly higher in the Pre + Post pups than in the Pre pups. A study with mice indicated that both deficient and enhanced autophagy leads to alterations in insulin sensitivity [13]. 

A previous study reported that the LC3-II:LC3-I ratio was low in the skeletal muscle of diabetic rats with high insulin levels [12]. This low LC3-II:LC3-I ratio indicates a reduced number of autophagosomes. In this study, we found a lower level of LC3-II protein in offspring with intrauterine protein restriction, but there was no significant difference in the LC3-II:LC3-I ratio among the three groups. 

Poor nutrition intake during pregnancy might change the hepatic autophagy regulatory mechanism even after the normalization of protein intake postnatally, considering that the levels of autophagy-related proteins remained low in 3-week-old Pre pups in the present study [24]. Low levels and ratios of LC3-I and LC3-II indicated persistent autophagic impairment in the Pre pups, despite the restoration of normal protein intake in the dams for 3 weeks.

Exposure to caloric restriction in the early postnatal period has been proven to increase hepatic autophagy following the restoration of a normal diet on day 21 in adult male offspring [5,22]. In the present study, the Pre + Post pups had lower protein levels of LC3-I and LC3-II, but an increased LC3-II:LC3-I ratio, suggesting that autophagy was activated, resulting in low levels of LC3-I and LC3-II. The Pre + Post pups did not experience a nutritional mismatch before or after birth, and the higher LC3-II:LC3-I ratio suggests a protective effect of maintaining a low protein diet postnatally. A previous study has reported that the restriction of energy intake increases autophagy activity, and this increase in autophagy activity is associated with insulin sensitivity and viability. In contrast, the LC3-II:LC3-I ratio did not significantly differ in the Pre pups that had a nutritional mismatch before and after birth in this study. As there was no significant difference in the autophagy level, the levels of stress markers in the skeletal muscle also did not significantly differ.

The skeletal muscle accounts for approximately 40% of the body’s mass and thus plays a major role in metabolic homeostasis. Autophagy deficiency in the skeletal muscle results in a marked decrease in fat mass, resistance to obesity, and improved insulin sensitivity. Muscle fiber affects the process of autophagy differently; the soleus muscle might have some effects on the process of autophagy, but the gastrocnemius muscle responds more sensitively to autophagy than the soleus muscle. This difference might be due to the differences between slow- and fast-twitch muscle fibers between the soleus muscle and gastrocnemius muscle. That is, the gastrocnemius muscle is more susceptible to external changes, such as starvation [25]. Here, we found no significant differences in the LC3-II:LC3-I ratios among the gastrocnemius, soleus, and plantaris muscles of the Pre and Pre + Post pups. This finding supports the notion that prolonged restriction might function as a postnatal therapeutic measure to reduce the risk of metabolic syndrome in adults [7]. 

Autophagy is a vital process for skeletal muscle maintenance and development. Muscle impairments contradictorily promote FGF21 secretion and insulin sensitivity in the entire body. Although not significant, the LC3-II:LC3-I ratio was lower in the Pre + Post group than in the Pre group, which suggests increased insulin sensitivity [13]. Western blotting showed that the rate of conversion from LC3-I to LC3-II, a key indicator of the autophagy process, was reduced, and that p62 accumulated in skeletal muscle-specific autophagy knockout mice, suggesting autophagy deficiency. p62 plays a role as an adapter to promote autophagy through association with ubiquitinated proteins. Thus, the p62 level increases when the autophagy process becomes active [19]. 

## 5. Conclusions

We conclude that maternal protein malnutrition suppresses autophagy at birth, which suggests autophagic impairment in neonatal rats with FGR. However, autophagic impairment persisted even though the body weight of suckling dams partially caught up with the restoration of normal diet for 3 weeks. The effects of autophagy differed between the liver and the skeletal muscle. Studies on the role of postnatal autophagy are scant, warranting further investigation.

## Figures and Tables

**Figure 1 nutrients-15-03058-f001:**
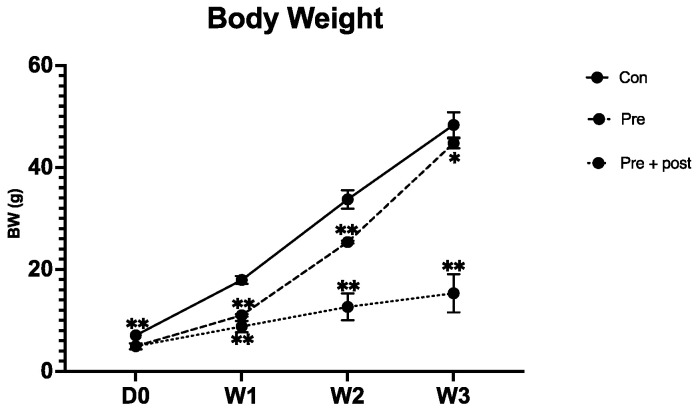
Body weight of pups on day 0 to week 3. Body weight of Con, Pre, and Pre + Post pups was recorded at birth and weeks 1, 2, and 3. Body weight of Pre and Pre + Post pups was significantly reduced compared with that of the Con group. Con, control; Pre, prenatal protein restriction; Pre + Post, prenatal and postnatal protein restriction. **, *p* < 0.0001; *, *p* < 0.05.

**Figure 2 nutrients-15-03058-f002:**
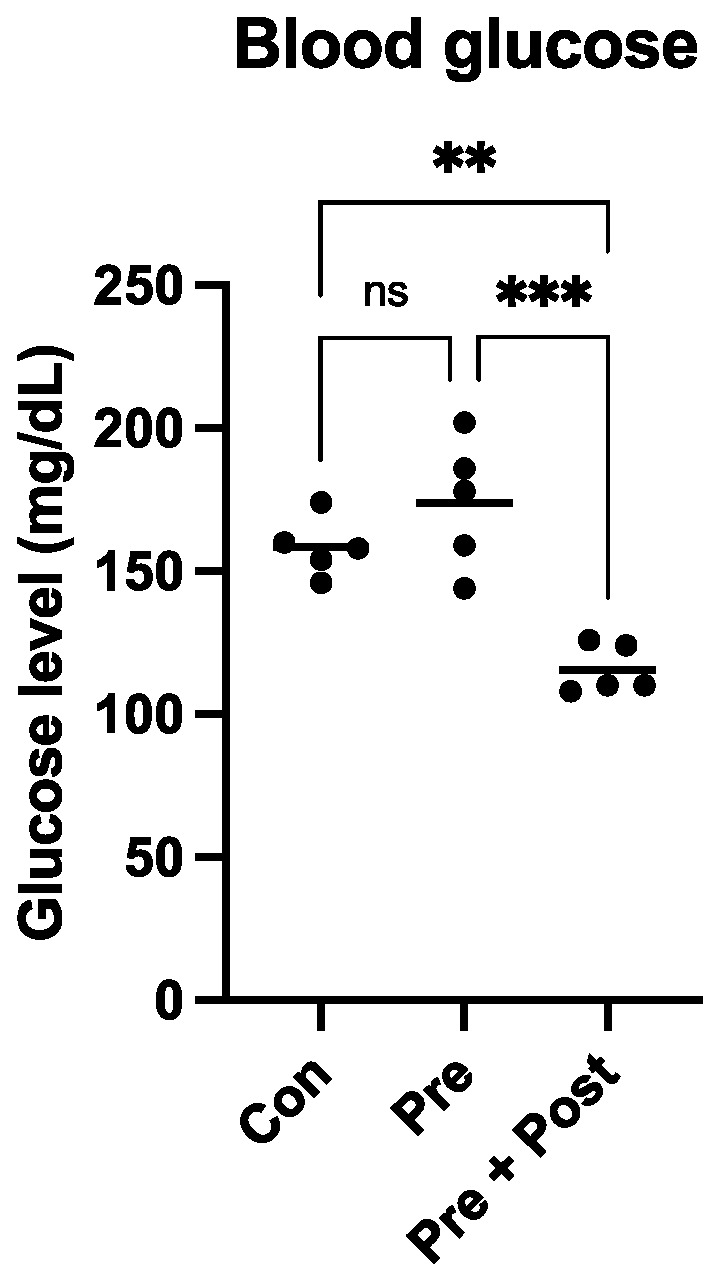
Blood glucose level on week 3. The blood glucose level was measured before organ sampling. Blood glucose levels between the Con and Pre groups were not significantly different. Con, control; ns, not significant; Pre, prenatal protein restriction; Pre + Post, prenatal and postnatal protein restriction. ***, *p* < 0.005; **, *p* < 0.05.

**Figure 3 nutrients-15-03058-f003:**
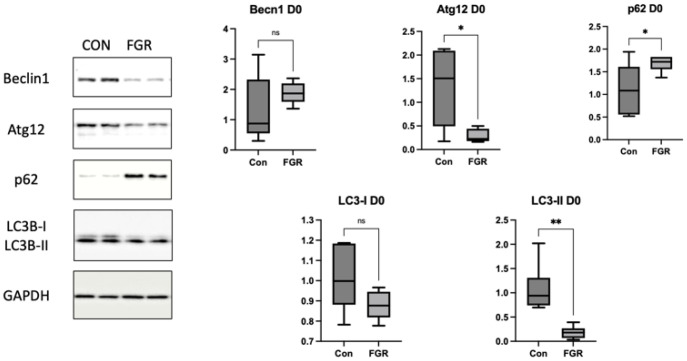
Western blot of the autophagic marker in livers of newborn pups (day 0). Protein expression of hepatic autophagy markers in newborn pups. Atg12 and LC3-II protein expression was significantly reduced in the Pre group compared to the Con group (*p* = 0.0015 and *p* = 0.01, respectively; *n* = 5 per group). Protein expression of the termination phase p62 significantly increased in the Pre group (*p* = 0.024). Bar columns represent the mean and error bars represent the standard deviation. Atg12, autophagy-related 12; Con, control; LC3-II, microtubule-associated protein light chain 3-II; ns, not significant; Pre, prenatal protein restriction. **, *p* < 0.005; *, *p* < 0.05.

**Figure 4 nutrients-15-03058-f004:**
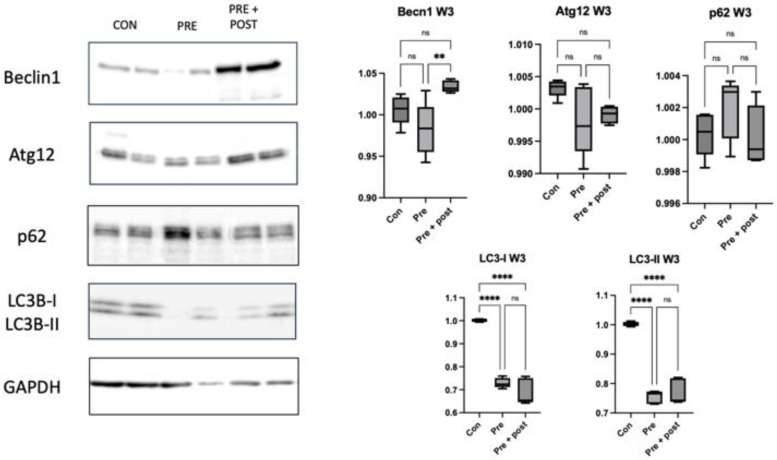
Western blot of the autophagic markers in livers of 3-week-old pups. Hepatic protein levels of LC3-I and LC3-II decreased in the Pre and Pre + Post groups compared with those in the Con group (*p* < 0.0001), whereas the level of beclin-1 increased in the 3-week-old Pre + Post pups compared with that in the Pre group pups (*p* = 0.0064), but it did not significantly differ from the levels in the Con group. Bar columns and error bars represent mean and standard deviation, respectively. Atg12, autophagy-related 12; Con, control; LC3-II, microtubule-associated protein light chain 3-II; ns, not significant; Pre, prenatal protein restriction; Pre + Post, prenatal and postnatal protein restriction. **, *p* < 0.005; ****, *p* < 0.0001.

**Figure 5 nutrients-15-03058-f005:**
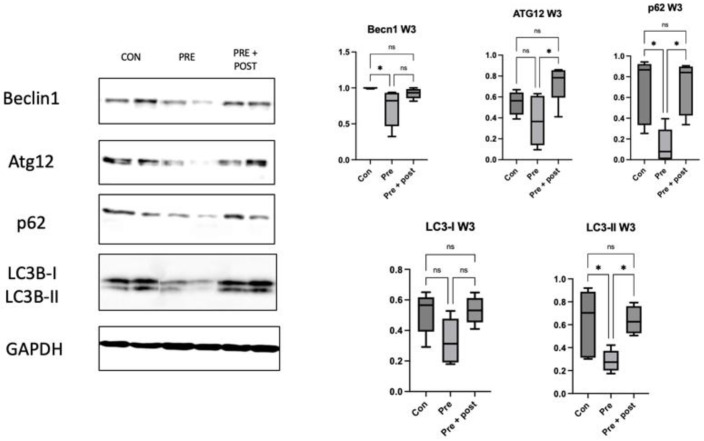
Western blot for skeletal muscle autophagic marker at week 3. Expression of autophagy regulator proteins beclin-1, LC3-II, and p62 decreased in the skeletal muscles of the Pre pups compared with that in the skeletal muscles of the Con pups (*p* = 0.04, *p* = 0.04, and *p* = 0.01, respectively). The levels of autophagy regulator proteins (Atg12, LC3-II, and p62) were also significantly lower in the Pre pups than in the Pre + Post pups (*p* = 0.02, *p* = 0.03, and *p* = 0.01, respectively). Atg12, autophagy-related 12; Con, control; LC3-II, microtubule-associated protein light chain 3-II; ns, not significant; Pre, prenatal protein restriction; Pre + Post, prenatal and postnatal protein restriction. *, *p* < 0.05.

**Figure 6 nutrients-15-03058-f006:**
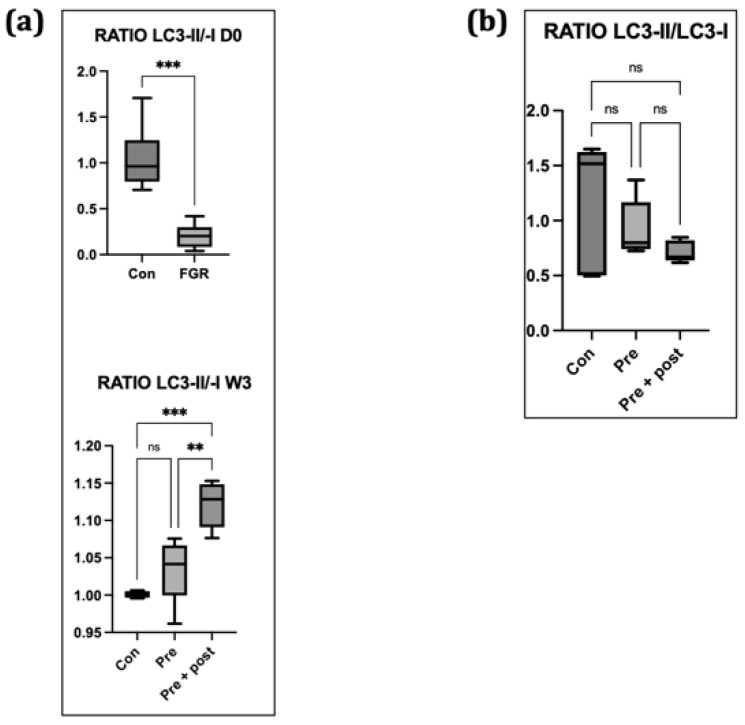
Ratio of LC3-II:LC3-I in the (**a**) liver and (**b**) skeletal muscles. In the liver, the ratio of LC3-II:LC3-I as autophagy markers was lower in Pre pups on day 0 and higher in Pre + Post pups than in the Pre and Con groups by week 3. Bar columns represent the mean, and error bars represent the standard deviation. Con, control; LC3-II, microtubule-associated protein light chain 3-II; ns, not significant; Pre, prenatal protein restriction; Pre + Post, prenatal and postnatal protein restriction. ***, *p* < 0.005; **, *p* < 0.05.

**Figure 7 nutrients-15-03058-f007:**
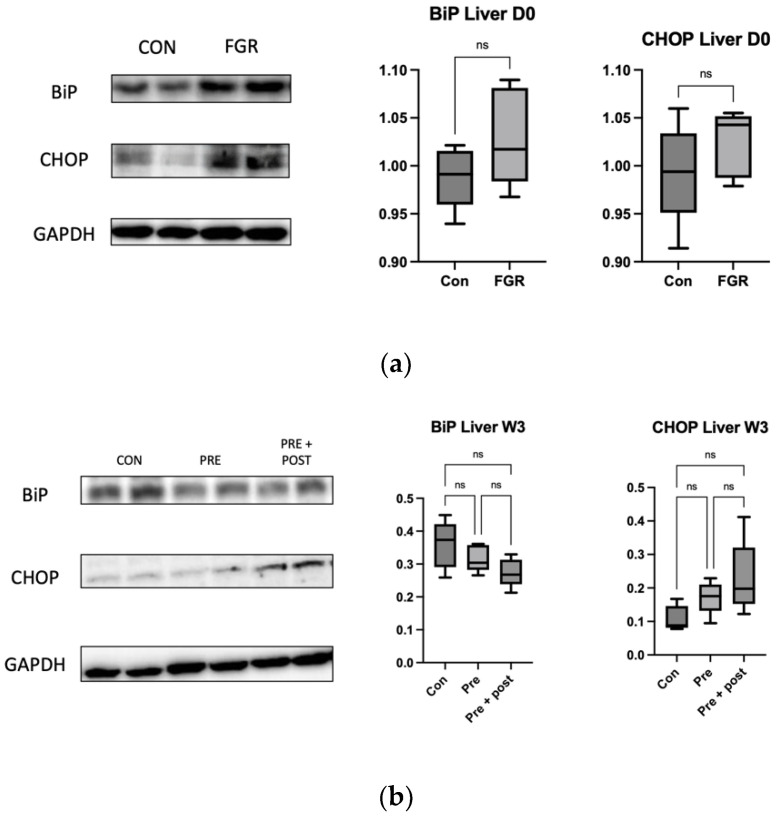
Western blot of the ER stress markers in the livers of (**a**) newborn pups and (**b**) 3-week-old pups. Hepatic CHOP and BiP protein levels in the Con and Pre groups were not significantly different in newborn pup livers. Bar columns and error bars represent mean and standard deviation, respectively. Con, control; FGR, fetal growth restriction; ns, not significant; Pre + Post, prenatal and postnatal protein restriction.

**Figure 8 nutrients-15-03058-f008:**
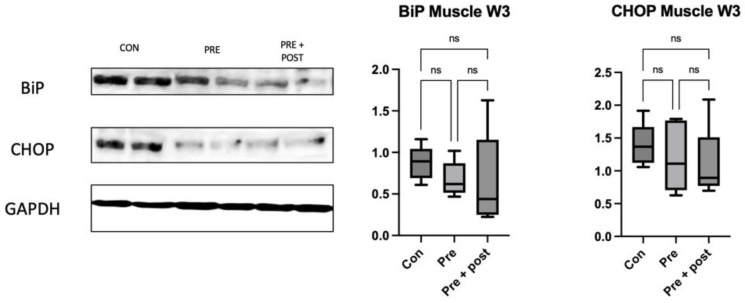
Western blot of skeletal muscle ER stress markers at week 3. Expression of ER stress marker proteins CHOP and BiP was not significantly different among the groups. Con, control; ns, not significant; Pre, prenatal protein restriction; Pre + Post, prenatal and postnatal protein restriction.

## Data Availability

Not applicable.

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
