# Peer review of "Hepatic and Skeletal Muscle Autophagy Marker Levels in Rat Models of Prenatal and Postnatal Protein Restriction"

_nutrients, 2023, doi:10.3390/nu15133058_

Round 1

Reviewer 1 Report

Santosha et al., have described the effects of protein restriction during prenatal and postnatal life on autophagy using a rat model.

Please if you could make clear some of the following points it would be helpful:

1) In the abstract first line suggests that calorie restriction during intrauterine and early postnatal growth ameliorates adult onset metabolic syndrome. But during the subsequent lines in the abstract have described adverse effects of protein restriction during intrauterine and early postnatal growth. If the author could explain the concepts better so that readers not aware of such restricted diet studies can understand. Is intrauterine and early postnatal protein restriction is same as a calorie restriction? Please explain in few lines in  the abstract or in the introduction part.

2) Lab animal studies are involve lots of time and effort to perform but I just wanted to know why the authors did not perform additional assays other western blot (immunofluorescence test or real time PCR assays?).

3) If you could reframe the sentences in the abstract wherein the autophagy markers are explained it could be helpful. For example increased P62 proteins can be mentioned along with its role concisely.

Author Response

Reponses to the Comments of the Reviewers   We thank you for the valuable comments and suggestions. We have revised the manuscript accordingly. Our responses to the comments are provided below.   1. In the abstract first line suggests that calorie restriction during intrauterine and early postnatal growth ameliorates adult onset metabolic syndrome. But during the subsequent lines in the abstract have described adverse effects of protein restriction during intrauterine and early postnatal growth. If the author could explain the concepts better so that readers not aware of such restricted diet studies can understand. Is intrauterine and early postnatal protein restriction is same as a calorie restriction? Please explain in few lines in  the abstract or in the introduction part.  We hypothesized that calorie restriction (including protein restriction) during intrauterine could cause adverse events such as metabolic syndrome. However, if the calorie (protein) restriction is continued until after the birth of offspring, the risk of adverse events will decrease. We have revised the relevant sentence as follows: “We postulated that FGR would impair, whereas IPPR would help regulate autophagy in neonatal rats.”   2. Lab animal studies are involve lots of time and effort to perform but I just wanted to know why the authors did not perform additional assays other western blot (immunofluorescence test or real time PCR assays?).  Animal research involves a long period, as you mentioned, and the size of Day-0 sample is too small to divide the tissues for two assays. Furthermore, a previous study (reference 10) stated “Changes in the amount of lapidated LC3 (LC3-II) on a western blot, could reflect a reduction in autophagosome turnover.” Therefore, we chose to perform western blot analysis.   3. If you could reframe the sentences in the abstract wherein the autophagy markers are explained it could be helpful. For example increased P62 proteins can be mentioned along with its role concisely.  We have added the following sentence for better understanding: “This finding suggests a lower hepatic autophagy flux in FGR offspring.”

Reviewer 2 Report

The study investigates the impact of fetal growth restriction (FGR) and prenatal and postnatal protein restriction (IPPR) on autophagy markers in neonatal rats.

The researchers aimed to understand the mechanism by which FGR and IPPR influence the risk of metabolic syndrome in adulthood. They compared autophagic markers in the liver and skeletal muscles of neonatal rats from three groups: a control group, FGR offspring group, and combined intrauterine and postnatal protein restriction (Pre+Post). Autophagy markers, including proteins like LC3-II, LC3-I, p62, and beclin 1, were assessed at birth and at the age of 21 days.

The results showed that the FGR groups had lower levels of autophagy-related proteins at birth, along with an increased level of p62 proteins and a lower LC3-II:LC3-I ratio. At 21 days of age, the hepatic levels of autophagy proteins were significantly decreased in the FGR and (Pre+Post) groups, but the LC3-II:LC3-I ratio was higher in the (Pre+Post ) group compared to the control and FGR groups. In the skeletal muscles, the protein levels of beclin 1, LC3-II, and p62 were lower in the FGR group, with no difference in the LC3-II:LC3-I ratio.

The findings suggest that maternal protein malnutrition suppresses autophagy in neonates, and a mismatch in the nutritional environment can affect downstream autophagic flux. The study highlights the potential role of autophagy in the developmental origins of metabolic diseases and insulin resistance.

The study presented is novel and exhibits scientific merit however, there are few major concerns that need to be addressed before considered for publication which are as follows:

1.      The image quality of all data sets needs to be improved as its pixelated and hard to comprehend the data and whether it matches the idea presented in the text.

2.      The authors need to show plasma glucose levels across the different groups to check on their insulin resistance state or other suitable markers to measure the same.

3.      Hepatic fat accumulation in these mice is not shown, this has to be included by performing oil-red-o staining on the liver sections.

4.      The authors need to stain for ER stress markers either by imaging to western blotting to add more support to their findings.

Author Response

Thank you for the insightful comments. We have carefully revised the manuscript accordingly. Our responses to the comments are provided below.   1. The image quality of all data sets needs to be improved as its pixelated and hard to comprehend the data and whether it matches the idea presented in the text.  We tried to improve the quality of the pictures and replaced Figures 2 to 5 with new western blot images.   2. The authors need to show plasma glucose levels across the different groups to check on their insulin resistance state or other suitable markers to measure the same.  We did not collect blood samples from the rats. However, we measured the glucose level with a glucose meter before euthanizing the rats, and the results have been added in the Results (page 4, line 161-163, Fig 2).   3. Hepatic fat accumulation in these mice is not shown, this has to be included by performing oil-red-o staining on the liver sections.  We do not have liver samples for Oil Red O staining. Instead, we planned to measure adipophilin expression using western blotting and placed an order for antibodies from a company outside Japan. As we could not obtain the antibodies before the deadline for resubmission, the relevant results could not be included in the manuscript.   4. The authors need to stain for ER stress markers either by imaging to western blotting to add more support to their findings.  We measured ER stress markers (BiP and CHOP) in the liver and skeletal muscle and added the results in the Results.

Reviewer 3 Report

Comments

Author Response

Thank you for the kind comment.